# The Influences of Different Mixing Methods for Fungi and Substrates on the Mechanical and Physicochemical Properties of Mycelium Composites

**DOI:** 10.3390/biology14040322

**Published:** 2025-03-22

**Authors:** Ziwei Dong, Dongyang Li, Yu Li, Shijun Xiao, Xuerong Han

**Affiliations:** 1Engineering Research Center of Chinese Ministry of Education for Edible and Medicinal Fungi, Changchun 130118, China; dongziwei@mails.jlau.edu.cn (Z.D.); lidongyang@mails.jlau.edu.cn (D.L.); yuli966@126.com (Y.L.); xiaoshijun@jlau.edu.cn (S.X.); 2College of Mycology, Jilin Agricultural University, Changchun 130118, China; 3Jilin Province Key Laboratory of MycoPhenomics, Jilin Agricultural University, Changchun 130118, China

**Keywords:** biocomposite, mycelium, cultivate

## Abstract

This study investigated mixing methods for optimizing mycelium-based packaging composites. We compared materials prepared through different fungal–substrate mixing methods in terms of appearance, mechanical strength, waterproof performance, thermal stability, and microscopic structure. The results demonstrated that pre-cultured mixtures achieved compressive strength ≥ 0.08 MPa and flexural strength ≥ 11 N. A microscopic analysis revealed improved mechanical properties from dense mycelium growth and effective substrate binding. The materials exhibited only 25% water absorption after 60 min, with surface mycelium forming a protective waterproof layer. All samples showed similar thermal degradation patterns (initial decomposition at 170 °C, peak at 350 °C), though volume reduction differed across mixing methods. This research offers practical guidelines for enhancing mycelium composite manufacturing processes.

## 1. Introduction

Mycelial composite material is an innovative material which skillfully combines fungal hyphae and agricultural waste [1,2]. With the significant advantages of a lack of pollution [3,4], easy degradation [5,6], and low cost [7,8], mycelium composites are gradually emerging in many scenarios and becoming widely used, especially in packaging [9], construction [10], home furnishings [11], etc. As for the standardization of the production, processing, and synthesis of mycelial composite materials, at present, researchers have not established a more systematic standard, but a basic production sequence has been established. However, there is little research on how to mix fungi and substrates, that is, the exploration of the first-stage culture method.

In the preparation of mycelial composites, the mixing methods of fungi and substrates are crucial and diverse [7,12]. For example, when using sawdust as a substrate to produce mycelial composites, the most common mixing method is to pretreat the sawdust, such as crushing it to the appropriate particle size and adjusting the humidity to a certain range, and then evenly spreading the selected fungal species, such as the spores or mycelial fragments of *Pleurotus ostreatus* mycelium, on the surface of the sawdust, and fully mixing it using mixing equipment to make the contact between the fungi and sawdust as even as possible so that the fungi can rapidly colonize and grow among the sawdust particles [13,14]. For another example, when the straw in agricultural waste is used as the substrate, the straw can be chopped into small pieces [15,16] and mixed with an appropriate amount of water to form a mixture with a certain humidity, and then, the mycelium of *Ganoderma lucidum* can be inoculated into the substrate [17,18,19]. The inoculation process can involve injecting the cultured *Ganoderma lucidum* mycelium liquid into a mixture of straw and water according to a certain proportion, and at the same time, incorporating them with manual overturning or mechanical stirring so that the mycelium is evenly distributed in the straw substrate, laying the foundation for the subsequent growth and spread of mycelium on the straw, and then forming mycelium composites with specific properties [20,21].

At present, there are some problems in the mixing methods for fungi and substrates: ① It is difficult to accurately control the mixing uniformity. Even at a relatively uniform macro level, local fungal concentration differences may still exist at the micro scale, which will lead to inconsistent mycelial growth rates in the subsequent cultivation process and affect the overall performance uniformity of the material, causing problems such as fluctuations in mechanical strength, density distribution, etc., which is not conducive to the production of high-quality and stable mycelial composites [22,23]. ② The impact of the mixing process on fungal activity. Some mixing methods involve mechanical force, ultrasound, static electricity, and other physical factors, or chemical reagent treatment, which may have a negative impact on the activity of fungi, thus affecting the formation efficiency and quality of the final mycelium composite. ③ The research on the adaptability of substrates to fungi is insufficient. At present, there is a lack of systematic and in-depth research on the optimal mixing mode between different kinds of fungi and various agricultural wastes or other substrates [24]. Different fungi have different requirements and preferences for the physical structure and chemical composition of the substrate, and the existing mixing methods often do not fully consider this suitability. At present, the development of relevant targeted mixing technology is insufficient, which leads to the ineffective utilization of some substrate resources.

In this study, through single-factor experiments, taking the mechanical properties, water absorption, chemical analysis as indicators, the influence of the mixing mode of fungi and the matrix on mycelial composites was explored. This study hopes to provide a reference for the preparation process optimization of mycelial composites.

*Ganoderma sichuanase* is one of the most commonly selected strains in the preparation process of mycelium-based composite materials [22]. In the preliminary pre-experiment, it was found that this strain has a high compatibility with soybean straw, which helps to enhance the stability of the connection between the mycelium and the substrate in the material. When considering industrial production in the future, it can be regarded as a fixed strain for production. Moreover, compared with other white-rot fungi such as *Pleurotus ostreatus* (0.04 MPa), *Pleurotus eryngii* (0.03 MPa), and *Lentinula edodes* (0.06 MPa), material prepared with this strain has a relatively high compressive strength (0.10 MPa) [23].

## 2. Materials and Methods

### 2.1. Materials

#### 2.1.1. Ganoderma Species

The strain G19 (*Ganoderma sichuanase*) used in this experiment was from Jilin Agricultural University, and PDA (Potato Dextrose Agar) medium was used for its activation and separation for subsequent experiments.

#### 2.1.2. Substrate Preparation

The substrate used in this experiment was soybean straw, which was sourced from the Mushroom and Vegetable Base of Jilin Agricultural University. Before the inoculation operation, it was necessary to prepare soybean straw and other additives according to specific proportions. Specifically, soybean straw accounted for 78%, wheat bran accounted for 20%, and lime and gypsum each accounted for 1%. The water content of the prepared mixed substrate had to maintained at around 60%.

### 2.2. Three Different Mixing Methods

The culture substrate formula used in this experiment consisted of soybean straw, wheat bran, lime, and gypsum in a ratio of 78:20:1:1, with a moisture content of 60%. Based on differences in substrate cultivation methods, the experiment was divided into three cultivation approaches: PDA fungal inoculation, pre-cultivation, and secondary inoculation. For the compression strength and water absorption tests, each group contained three parallel samples. Each sample had a wet-material weight of 150 g (mold dimensions: 95 mm × 60 mm × 50 mm). For the bending strength tests, each group included five parallel samples, with each sample weighing 50 g (mold dimensions: 140 mm × 20 mm × 25 mm). The preparation process of mycelium-based packaging composites was divided into two main stages. The core objective of this study was to explore optimization strategies for different cultivation methods during the first stage, specifically adopting three distinct cultivation approaches as follows:

#### 2.2.1. Fungal Inoculation (FI)

The prepared substrate was uniformly dispensed into pre-sterilized molds and compacted using a flattening tool to achieve surface consolidation. Following this, the substrate underwent sterilization in an autoclave (GR85DA, Zealway Instrument Inc., Xiamen, China) at 121 °C for 60 min, followed by cooling to ambient temperature within a laminar flow hood. Under aseptic conditions, mycelia pre-cultured on PDA plates were aseptically divided into equal portions, each of which was inoculated into a separate mold. The sealed molds were subsequently transferred to an incubation chamber (GZX-300BSH-III, Shanghai Cimo Medical Instrument Co., Ltd., Shanghai, China) maintained at 25 °C with controlled humidity and light exclusion, where they remained until complete mycelial colonization of the substrate was achieved.

#### 2.2.2. Pre-Culture (PC)

Activated PDA strains were inoculated into pre-formulated substrates contained within sterilized spawn bags. Upon achieving complete mycelial colonization of the substrate, the resultant spawn blocks were aseptically fragmented in a laminar flow hood. This mechanical disruption process converted the fully colonized substrate into granular material with relatively uniform particle sizes. The fragmented substrate was then transferred into pre-sterilized molds. Following assembly, the sealed molds were incubated in light-protected conditions at 25 °C for 3 days within a temperature-controlled cultivation chamber (GZX-300BSH-III).

#### 2.2.3. Secondary Inoculation (SI)

Activated PDA strains were aseptically inoculated into pre-formulated substrates contained within sterilized spawn bags (also termed fungal cultivation blocks). Following complete mycelial colonization of the substrate, the spawn blocks were stored at 4 °C for subsequent experimental use. The prepared substrate was loaded into spawn bags and subjected to a sterilization protocol (121 °C, 60 min). Concurrently, all experimental apparatus—including molds, scissors, forceps, containers, and flattening tools—underwent identical sterilization conditions to ensure aseptic operational environments and eliminate contamination risks. Post-sterilization, materials were cooled to ambient temperature within a laminar flow hood. An alcohol lamp was ignited to maintain an aseptic workspace. Based on wet-weight measurements, 10–32% of pre-cultured mycelium–substrate mixture was precisely weighed and homogenized with fresh substrate to ensure a uniform compositional distribution. The blended substrate was transferred into molds, compacted using a flattening tool, and hermetically sealed with identification labels. Finally, the sealed molds were incubated in light-protected conditions at 25 °C for 3 days within a temperature-controlled cultivation chamber (GZX-300BSH-III).

#### 2.2.4. Material Post-Processing

The second stage of preparing the mycelial composite materials was material post-treatment: after the mycelial system of each treatment group was fully constructed and the mycelium was covered with the substrate, the sample was demolded using the tin foil attached to the mold and placed in a slightly larger sealed container. It was further cultured in the same environment for 3 days to ensure that the mycelial surface in contact with the mold wall continued to grow and form a complete, dense, and white sample. Subsequently, the material sample was placed in a 60 °C oven (DHG-9140, Shanghai Yiheng Scientific Instrument Co., Ltd., Shanghai, China) and dried to a constant weight, and the sample state was adjusted at 25 °C for subsequent experiments.

### 2.3. Performance Operation

#### 2.3.1. Scanning Electron Microscopy (SEM)

Specimen collection was performed by selecting both core and surface regions of fungal-incubated substrates. The samples were dehydrated in a drying oven (60 °C, 24 h), then mechanically homogenized through a 60-mesh sieve. Following gold sputter coating (5 nm thickness, 20 mA, 120 s), morphological characterization was conducted using scanning electron microscopy (Sigma 500, Carl Zeiss AG, Jena, Germany) under high-vacuum mode (10^−3^ Pa) with an accelerating voltage of 10 kV at working distances of 8–10 mm. Image acquisition was executed at 130× and 500× magnification with secondary electron detection mode.

#### 2.3.2. Density and Shrinkage Rate

Following dehydration of the mycelium composite specimens to a constant mass under controlled conditions (60 °C, 72 h), bulk density determination was conducted in strict compliance with Chinese National Standard [25] (Testing methods for density of composite materials). Shrinkage rate (1) was systematically performed at two critical stages: (1) a post-processing stage including secondary mycelial development and structural consolidation, and (2) post-thermal treatment conditioning (24 h equilibration period at 25 °C ± 1 °C, 50% RH). All metrological procedures were executed using ISO-certified instrumentation with measurement uncertainty maintained below 0.5% of the full scale.
(1)$$\mathrm{Shrinkage}\;\mathrm{rate}=\frac{V1-V2}{V1}\times 100\%$$

*V*1: Before drying; *V*2: after drying.

#### 2.3.3. Mechanical Properties 

The compression testing was conducted using a computer-controlled universal testing machine (WDW-5A, Jinan Wenteng Test Instrument Co., Ltd., Jinan, China) in strict compliance with the national standard [26].Rigid cellular plastics—Determination of compression properties). Specimens with nominal dimensions of 100 mm (L) × 60 mm (W) × 50 mm (H) (±0.5 mm tolerance) were conditioned at 23 ± 2 °C and 50% RH for 48 h prior to testing. The quasi-static loading protocol employed a constant crosshead speed of 10 ± 2 mm/min. Specimen failure was defined as either (a) the attainment of 10% engineering strain or (b) the onset of macroscopic surface cracking, whichever occurred first.

Flexural characterization followed the national standard [27]. Displacement-controlled loading at 1 mm/min was applied until meeting either of the following termination criteria was met: (a) a 20% deflection-to-span ratio or (b) complete structural failure.

All measurements were performed in triplicate with a <2.5% coefficient of variation between replicates.

#### 2.3.4. Water Absorption Performance 

Water absorption behavior was quantified following the national standard [28]. The test specimens (*n* = 3 per group) underwent controlled immersion in distilled water maintained at 23 ± 2 °C. Following initial immersion (10.0 ± 0.5 s), the specimens were retrieved and subjected to gravitational drainage on a steel mesh (3.00 ± 0.05 mm aperture) positioned at a 30° ± 1° inclination relative to the vertical plane for a 30.0 ± 0.5 s drainage duration, and we recorded the material mass *m*_0_ at this time. Subsequent cyclic immersion phases (1, 30, and 60 min durations) employed identical drainage parameters, with the saturated mass (*m_n_*) recorded at each interval. The water absorption ratio (WAR) was calculated using Equation (2):(2)$$\mathrm{WAR}\;(\%)=\frac{m_{n}-m_{0}}{m_{0}}\times 100$$

#### 2.3.5. FT-IR 

Approximately 1~2 mg of sample powder was precisely weighed and homogenized with pre-dried potassium bromide (KBr) powder at a mass ratio of 1:100. The mixture was thoroughly ground in an agate mortar until a uniform particle size of approximately 2 μm was achieved. The finely ground powder was subsequently transferred into a pellet die and compressed under a pressure of 15 MPa to form a transparent or semi-transparent pellet. Fourier transform infrared (L1600400, PerkinElmer Inc., Waltham, MA, USA) spectroscopy was performed using a spectrometer configured with a resolution of 4 cm^−1^, 32 cumulative scans, and a wavenumber range of 4000~500 cm^−1^. The spectral data were collected after completion of the measurement for further analysis.

#### 2.3.6. Thermogravimetric

Approximately 10.0 ± 0.2 mg loaded into platinum crucibles within a thermogravimetric analyzer (TGA 4000, PerkinElmer Inc., USA). The system was purged with high-purity nitrogen (99.999%) at a flow rate of 50.0 ± 0.5 mL/min for 30 min prior to heating to establish dynamic inert atmosphere conditions. Temperature programming was executed under dynamic nitrogen atmosphere (99.999% purity) at a constant flow rate of 50.0 ± 0.5 mL/min, implementing a linear heating regime from 30 °C to 600 °C at β = 10.0 ± 0.1 °C/min. Isothermal conditioning at 600 °C was maintained for 10 min to ensure complete thermal equilibration prior to data acquisition. Triplicate measurements demonstrated mass variation within ±0.25% across replicates. 

#### 2.3.7. Statistical Analysis

Data processing was performed using Microsoft Excel 2010, with statistical analysis conducted through SPSS Statistics 2019, International Business Machines Corporation, Armonk, NY, USA. One-way analysis of variance (ANOVA) was employed for significance testing, adopting α = 0.05 as the statistical significance threshold (*p* < 0.05). Graphical representations were generated using OriginPro 2021 (Version 9.8.5.612).

## 3. Results

### 3.1. Morphological Analysis

The differences in the appearance of materials under different mixing methods are mainly reflected in the state after the establishment of the mycelium system (Figure 1). Firstly, the materials obtained by the three mixing methods are all white in color, with a smooth surface, no obvious bulging or shrinkage deformation, good fusion, no obvious particle shedding phenomena, and no obvious stains or impurities. Immediately after demolding, the growth states of the mycelium vary due to different inoculation methods. As can be seen from the mycelium coverage degree on the surface of the samples in Figure 1, compared with the other two mixing methods, the mycelium on the surface of the samples obtained by the SI method is mostly aerial mycelium and has not formed a relatively dense mycelium membrane. The main reason is that during inoculation, the strains are dispersed in the substrate and are not completely and evenly mixed with the substrate. As a result, under the same cultivation time, the growth of individual samples is uneven in the later stage. Through visual observation after the samples are demolded, it is found that since the oxygen content in the external area is relatively sufficient compared with that in the mold, the surface of the samples is completely colonized on the second day after demolding, and the mycelium on the surface of the samples is white and grows evenly. Three days after the secondary culture, a layer of white and dense fungal skin is formed on the surface of the samples in the FI and PC groups. However, a complete fungal skin is not formed on the surface of the materials obtained by SI, and there is more aerial mycelium on the surface. It is speculated that the cultivation time in the mold is relatively short, and the mycelium has not formed a good network structure with the substrate, so the surface fungal skin is incomplete during the secondary culture compared with the former two.

It can be seen in Figure 2a that the mycelium is tightly colonized on the surface of the substrate and its hyphae are intertwined with each other. The positions where the mycelium and the substrate are combined are decomposed into small granular substances. Following biological pre-cultivation, the PC group specimens were mechanically disintegrated and re-compacted into molds. Each mycelium-encapsulated substrate particle demonstrated rapid hyphal proliferation, with re-established interparticle connections forming a consolidated mycelial network architecture during secondary consolidation. As Figure 2b shows, the mycelium is relatively abundant compared with the other two mixing methods, densely wrapping the substrate. In the case of SI, after the prepared and sterilized substrate is mixed with a certain proportion of secondary spawn, there are growth and germination points for the mycelium everywhere in the material, thus establishing a network combining the mycelium and the substrate. However, since it is manually mixed, it cannot be ensured that every corner has substrate with mycelium. Therefore, the combination situation of the mycelium and the substrate in the material is slightly different from that of the other two methods, but the mycelium network is basically same, with the mycelium covering the surface of the substrate and decomposing the substrate as well. Figure 2d presents the microscopic morphology of the fungal skin film formed on the surface of the material. A large number of mycelium hyphae are piled together and exhibit the phenomenon of entanglement. The mycelium grows layer by layer to establish a three-dimensional network structure, and the gaps between the mycelium hyphae are almost filled by the mycelium, meaning that the surface of the mycelium composite is more complete. To summarize, there are slight differences in aspects such as the amount of mycelium and the combination mode of the mycelium and the substrate among the materials prepared under different mixing methods. The amount of mycelium in the PC group is the most abundant, while that in the FI and SI groups is slightly less abundant, and the exposed substrate can be seen. The amount of mycelium and the combination mode of the mycelium and the substrate also affect the mechanical properties of the materials [29,30].

### 3.2. Density and Shrinkage Rate Analysis

The density range of the three mixing methods is 0.16~0.17 g/cm^3^, with no significant differences (Table 1). The density of mycelium composite materials is related to the selected substrates. In the early preliminary experiments, it was found that the density of mycelium composite materials prepared with cottonseed hulls was around 0.3 g/cm^3^, while that of samples prepared with corn stalks was around 0.14 g/cm^3^. The average density of the mycelium composite materials prepared in this study is above 0.16 g/cm^3^, which is low when compared with materials such as polystyrene (11~50 kg/m^3^), polyurethane (30~100 kg/m^3^), and phenolic resin foam (35~120 kg/m^3^). Through the analysis of density and elastic modulus values in the Ashby plot, the mycelium-based packaging composite material was identified as falling within the “foam” category. This is relevant considering that when mycelium composite materials are applied to packaging in the future, people will often take into account the economic cost of logistics and transportation.

The shrinkage rate of SI among the three mixing methods is about 25%. Because the mycelium composite material is mainly composed of agricultural waste and mycelium, about 60% of the water in it will cause volume shrinkage when drying when preparing the substrate, and under high-temperature conditions, the mycelium begins to inactivate, the water existing in the cell wall evaporates, and the mycelium shrinks, which is also one of the reasons for material shrinkage.

The quantitative analysis of density-shortening rate correlation reveals a statistically significant interdependence, wherein the shrinkage rate constitutes a critical determinant of material densification. This causal relationship arises from the density calculation protocol: material density is calculated based on post-dehydration volume using a formula where an increasing shrinkage rate inherently elevates through progressive volume reduction. As a promising alternative to expanded polystyrene (EPS) foams, mycelium-based packaging composites currently exhibit elevated shrinkage rates (20.83 ± 0.99% vs. EPS < 5%), presenting a critical challenge for industrial-scale manufacturing standardization. Mitigation of this dimensional instability requires the optimization of mycelial network architecture and substrate-binding kinetics during the consolidation phase.

### 3.3. Mechanical Properties Analysis

Compressive strength is one of the important indicators of material mechanical properties, and is reflected by a stress–strain curve. The compressive strength of the material will gradually increase with an increase in strain (Figure 3a). All curves show an upward trend with an increase in strain, and there is no peak. Under strain of 10%, the stress that the FI group and PC group can bear is greater than that of the SI Group, reaching more than 0.08 MPa. In the three mixing methods, the SI group had a shorter culture time than the other two, and the mold fell off just after the hyphae and the substrate were established, so the hyphae and the substrate were not tightly combined, resulting in a lower compression strength than for the other two mixing methods. The mycelial composite prepared by Rom’an Ramos [31] has compressive strength in the range of 20–200 KPa under 10% strain.

The flexural strength is mainly reflected by the fracture bending load. Figure 3c shows the fracture bending load under three different mixing methods. Both the FI and PC groups are above 11 N, which is higher than the minimum requirement (11 N). The average breaking bending load of the SI group is only 7.94 N, while that of the FI group is more than 18 N. Flexural strength mainly reflects the flexural strength of the hyphae and substrate inside the material. The type of strain, the type of substrate, and the degree of hypha-and-substrate binding are important factors affecting performance. During the test, it was found that a layer of dense mycelial film formed on the surface of the material played an important role in protecting the material from bending.

### 3.4. Water Absorption Performance Analysis

The initial water immersion testing (1 min duration) revealed comparable water absorption rates across all three cultivation methods(Figure 4), ranging between 2 and 3%. However, divergent trends emerged with prolonged immersion: at 30 min, PC-group specimens exhibited moderate absorption (11%), whereas SI- and FI-group samples demonstrated significantly higher uptake exceeding 20%. At 60 min, absorption disparities intensified, with the PC group reaching 25%, contrasted by the SI group (42%) and FI group (35%).

Distinct morphological variations were observed among specimens from different cultivation methods following water immersion. Initial immersion trials (1 min and 30 min) showed negligible surface alterations across all groups compared to the pre-immersion baselines. At 60 min of immersion, progressive interfacial degradation became evident (Figure 4): the FI group maintained structural integrity with minor yellowing discoloration of the mycelial membrane and limited substrate exposure; the PC group exhibited moderate soybean stalk substrate exposure (12–15% surface area) and retained >90% mycelial membrane coverage; the SI-group specimens demonstrated the poorest water resistance among the three mixing methods, with dual surface deterioration phenomena emerging after 60 min of immersion, including yellowing discoloration and localized substrate exposure. This was attributed to incomplete mycelial membrane formation during secondary cultivation, where superficial aerial hyphae dissolved upon hydration [32].

Notably, all specimens maintained macroscopic structural integrity without disintegration or fragmentation. A cross-sectional analysis revealed limited water penetration depth, confined to peripheral regions, with central zones remaining unaffected. The post-dehydration mass recovery rates reached 98.7 ± 0.5% of the original values, confirming reversible hygroscopic behavior.

### 3.5. FT-IR Analysis

The FTIR spectra of mycelial composites prepared by different mixing methods are shown in Figure 5. In addition to starch and protein in wheat bran, the matrix is also composed of different amounts of lignin, cellulose, hemicellulose, and other macromolecules. In addition to the above compounds, chitin, protein, and β-glucan are also components of the fungal cell wall in the three mixing methods, unlike in the substrate samples without inoculation of fungi [33,34].

The peak at 3045 cm^−1^ represents the stretching vibration of -OH in the hydroxyl group. This -OH comes, on one hand, from H_2_O, and on the other hand, from the alcohols and phenols in cellulose and lignin [35,36]. Compared with the substrate without inoculated strains, the other three samples contain mycelia, which hold water, thus increasing the -OH content.

Both the substrate inoculated with fungi and the substrate not inoculated with fungi show C-H vibration at 2920 cm^−1^, which is characteristic of lignin [35,36]. At 1650 cm^−1^, C=C stretching occurs on the lignin side face; in addition, α-chitin exists in the skeleton hyphae [37], β-chitin mainly exists in the reproductive hyphae, and the strain used in this experiment is *Ganoderma* spp., which has skeleton hyphae, so it is presumed that this peak may also be α-chitin [38]. The spectral generation between 1650 cm^−1^ and 1245 cm^−1^ is mainly attributed to lignin.

This analysis shows that the chemical composition has a secondary effect on the mechanical properties, and the mechanical properties of mycelium composites are mainly related to the microstructure (the interaction between mycelium and the substrate and the surface mycelium membrane). The change in the peak values of the four bands at the hydroxyl group in the Figure 5 helps to explain the degradation of polysaccharides (cellulose and hemicellulose) and water evaporation (the hydroxyl group is weak, and the hyphal content is low, which leads to a reduction in compression strength).

### 3.6. Thermogravimetric Analysis

The mycelial composites prepared by different mixing methods showed similar thermal distribution, the results are shown in Figure 6, and the thermal degradation was divided into three stages: 

The first stage is from the initial temperature to 172 °C. In the initial stage, with the increase in temperature, the material performance is unstable, and about 8% to 10% mass loss begins to occur after heating. This stage mainly involves the evaporation of water, which is mainly the evaporation of free water in the material and the loss of bound water in the biomass cell wall [37]. The bound water in the polymer will only evaporate when the temperature is greater than 100 °C.

The second stage is 172 °C to 428 °C, which mainly corresponds to the degradation of macromolecular components in mycelial composites. It mainly includes the polymerization and decomposition of chitin, amino acids, lipids, and polysaccharides, as well as the acetylation and deacetylation units of chitin in mycelium [38]. These polymers are mainly degraded by decomposition into small molecules. When the temperature exceeds 250 °C, the mass of the materials prepared by the three mixing methods begins to reduce considerably because of the rapid degradation of carbohydrates such as cellulose and hemicellulose. At this time, a smaller peak appears at about 300 °C in the DTG curve, which is due to the degradation of hemicellulose; then, the peak value of the curve reaches its maximum value at 350 °C, corresponding to the degradation of cellulose, which has a higher degradation temperature.

The third stage occurs after 400 °C, the TGA curve and DTG curve gradually flatten, and the wet weight of mycelial composite material decreases. This stage mainly involves the decomposition of lignin [39]. Due to its molecular complexity, lignin decomposes slowly at a high temperature of 190~900 °C, and it is difficult to degrade the benzene ring [24,36]. A study on extracting chitin from *Agaricus bisporus* showed that the mass loss was about 63% in the temperature range of 200~400 °C [40,41], while for *Ganoderma* species, the peak degradation temperature was 313 °C, and it could be completely degraded in the range of 400~500 °C.

The thermal degradation efficiency of different mixing methods began to differ after 350 °C, which may be related to the amount of residual nutrients in the substrate after mycelium degradation in different culture methods. The maximum thermal degradation rate was 0.7%/min. When the temperature exceeded 600 °C, the residual nutrients in FI and SI groups were higher than those in PC group. In the PC group, the mycelium content in the mycelium composite was higher, and the interaction between mycelium and the substrate was more intense, but at the same time, the thermal stability was reduced.

## 4. Discussion

By investigating different mixing protocols of fungal–mycelium composites, this study demonstrated that variations in mixing methods significantly influenced material properties, with pronounced differences observed in appearance, mechanical strength, and water absorption capacity.

Mechanical properties serve as critical evaluation metrics for mycelium-based composites. The most pronounced distinction among the FI, PC, and SI groups during cultivation lies in the varied cultivation durations caused by differences in fungal–substrate mixing protocols. Cultivation time directly governs mycelial biomass content and viability, which, in turn, determines the mechanical performance of the composites. Sun et al. [42] postulated that extended cultivation allows enhanced mycelial colonization within the substrate, forming an interconnected three-dimensional adhesive network that strengthens interfacial bonding—a finding consistent with our results. Scanning electron microscopy (SEM) further corroborated the critical role of mycelium–substrate interactions in mechanical enhancement.

Moreover, post-demolding secondary cultivation emerged as a decisive step: it promotes the formation of a dense mycelial membrane on the material surface, which significantly contributed to mechanical reinforcement in our experiments. Notably, the PC and FI groups exhibited prolonged cultivation periods compared to the SI group, enabling a mycelial transition from vegetative to reproductive growth. During secondary cultivation, these groups achieved maximal hyphal tensile strength alongside observable lignification and primordia formation. In contrast, the SI group’s shorter growth cycle retained hyphae predominantly in the vegetative phase with abundant aerial mycelia, resulting in fragmented surface membranes. This critical process remains underreported in the existing literature on fabrication protocols.

A critical limitation hindering the practical application of mycelium-based composites lies in their poor water resistance. In this study, the developed mycelium packaging composites exhibited significantly higher water absorption than conventional commercial foams, particularly after 60 min of immersion. This phenomenon stems from the abundant lignocellulosic components in the raw materials, which inherently display hydrophilicity. The cellulose cell walls contain numerous O-H bonds that form new hydrogen bonds with water molecules through interactions between H-groups of water and O-H groups in the fibrous substrate. These molecular-scale interactions induce intralamellar swelling, thereby amplifying water uptake [43]. Beyond strategies utilizing particulate substrates and compression molding [44], emerging approaches such as bio-based coatings [45] (e.g., polyfurfuryl alcohol resin, PFA) have demonstrated efficacy in reducing water absorption in natural fiber composites. Such methodologies could be strategically adapted to enhance the hydrophobicity of mycelium composites, addressing their current limitations in moisture-sensitive environments.

The practical application of mycelium-based packaging composites is further constrained by parameters such as density and shrinkage rate. While this study primarily employed density, shrinkage rate, and water absorption capacity as screening criteria for optimizing mixing protocols, these factors—though not exhaustively analyzed for their real-world performance limitations—remain critical research entry points to advance the industrial-scale application of mycelium composites, warranting systematic investigation.

## 5. Conclusions

This study investigated the effects of fungal–substrate mixing protocols on mycelium-based composites using soybean straw inoculated with *Ganoderma sichuanense* (G1) as the substrate. Material performance was systematically evaluated through a single-factor experimental design, with the key metrics including the morphological characteristics, compressive strength, flexural strength, density, shrinkage rate, water absorption capacity, Fourier transform infrared (FTIR) spectroscopy, and thermal properties. The experimental results revealed that mixing method variations induced prolonged cultivation periods, which significantly enhanced mycelial biomass and hyphae–substrate interactions, thereby generating substantial performance disparities—particularly in the mechanical properties. Among all of the protocols, the pre-cultured (PC) group demonstrated optimal fungal–substrate compatibility, yielding composites with superior integrated performance. Notably, critical parameters such as the water absorption and shrinkage rate observed in this system merit further investigation to advance functional optimization.

## Figures and Tables

**Figure 1 biology-14-00322-f001:**
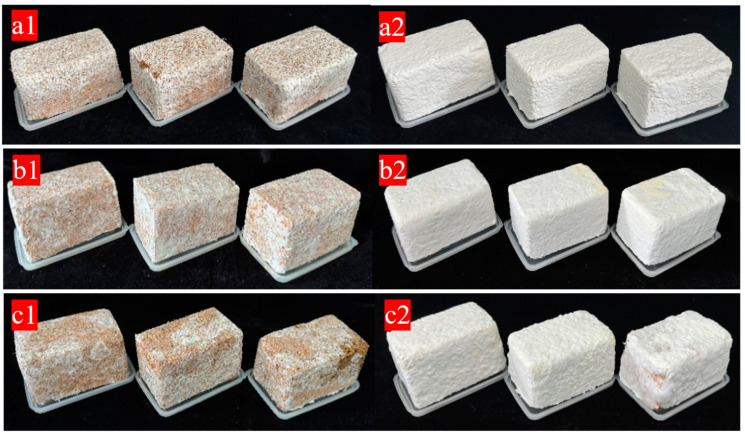
(**a1**,**b1**,**c1**) FI, PC, and SI immediately after being demolded; (**a2**,**b2**,**c2**) FI, PC, and SI after being cultured for three days.

**Figure 2 biology-14-00322-f002:**
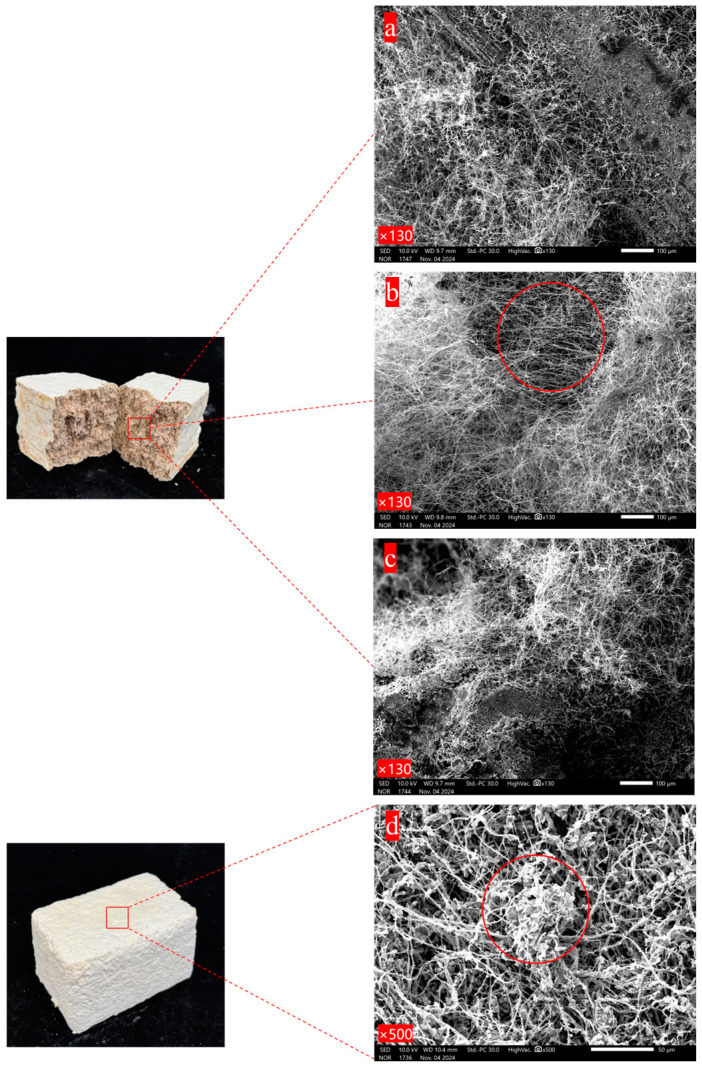
Scanning electron micrograph of different mixing methods: (**a**) FI; (**b**) PC; (**c**) SI; (**d**) material surface.

**Figure 3 biology-14-00322-f003:**
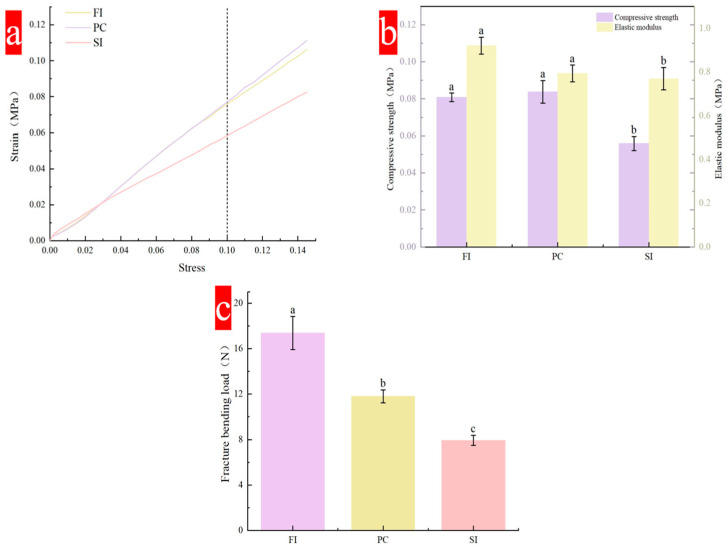
Mechanical properties of composites under different mixing methods: (**a**) stress–strain curves; (**b**) compressive strength and elastic modulus; (**c**) flexural strength. Different lowercase letters indicate that the different mixing methods are significantly different (*p* < 0.05).

**Figure 4 biology-14-00322-f004:**
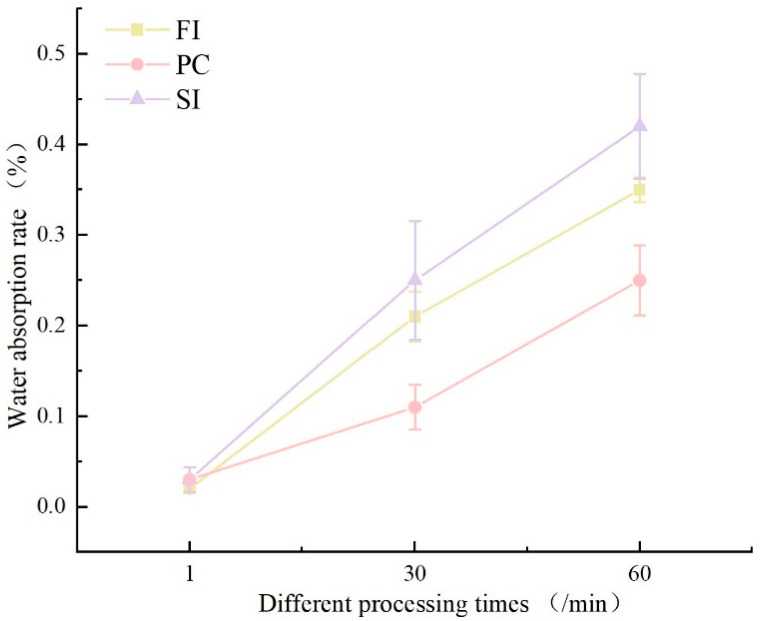
The water absorption rates under different mixing methods.

**Figure 5 biology-14-00322-f005:**
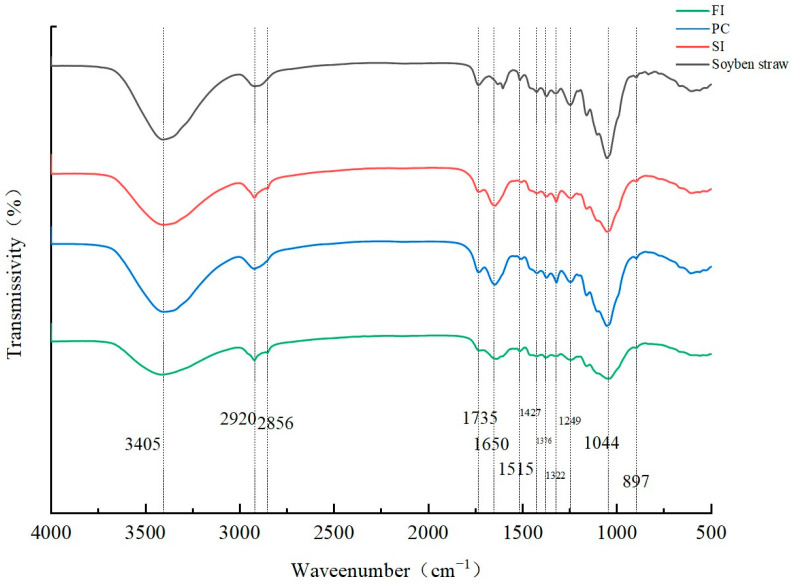
FT-IR spectra under different mixing methods.

**Figure 6 biology-14-00322-f006:**
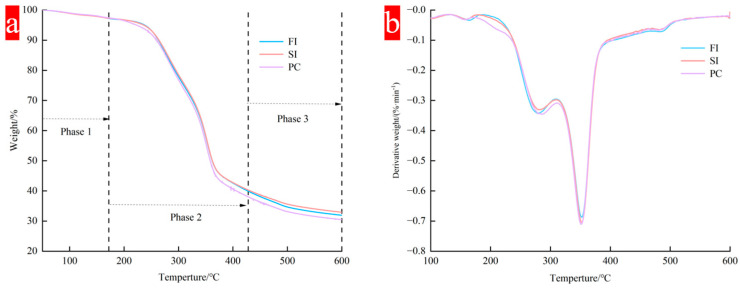
Thermogravimetric analysis of different mixing methods; (**a**) thermogravimetric analysis (TGA); (**b**) derivative thermogravimetry (DTG).

**Table 1 biology-14-00322-t001:** The shrinkage rate and density with different mixing methods.

Mixed Method	Shrinkage Rate (%)	Density (g/cm^3^)
FI	20.83 ± 0.99 ^b^	0.1648 ± 0.0040 ^b^
PC	24.79 ± 0.96 ^a^	0.1792 ± 0.0092 ^a^
SI	25.39 ± 1.68 ^a^	0.1797 ± 0.0028 ^a^

Note: The data are presented as $\bar{X}\;\pm \;s$. Different lowercase letters indicate that the different mixing methods are significantly different (*p* < 0.05).

## Data Availability

All data obtained during this research are included in the article.

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
