# Peer review of "The Influences of Different Mixing Methods for Fungi and Substrates on the Mechanical and Physicochemical Properties of Mycelium Composites"

_biology, 2025, doi:10.3390/biology14040322_

Round 1
Reviewer 1 Report
Comments and Suggestions for Authors
This article investigates the effects of different fungal-substrate mixing methods on mycelium composite properties, which holds reference value for both researchers and readers. However, the writing appears immature, with unclear prioritization in methods,results,analysis and notable formatting errors. Significant improvements are required before publication in this journal. The detailed recommendations are as follows:
- The title of the article lacks conciseness, and it is recommended to revise it.
- Table 1 contains unit labeling errors. Systematically verify measurement units in all tables.
- Figures and tables fail to meet academic standards (incomplete captions, insufficient logical coherence).
- Redundant descriptions in experimental procedures (e.g., Section 2.2). Streamline language and supplement critical parameters:
- Ambiguous demarcation between two material preparation stages. Suggest separate subsections.
- Rename "Chemical Analysis" subsection to "FTIR Analysis" for accuracy.
- Optical microscopy mentioned in morphological analysis lacks corresponding result images. Add relevant images or remove unrelated method descriptions.
- Morphological Analysis (e.g., Section 3.1): Current optical micrographs only show single samples - insufficient for comparing three mixing methods.
Descriptions deviate from core arguments. Add comparative micrographs of
three sample groups and correlate morphological features with material
properties.
- Water absorption discussion (3.4) contains divergent arguments. Focus on key data and remove redundant descriptions.
- Weak correlation between volume loss and density analysis (3.2). If merging discussions, supplement evidence clarifying their intrinsic relationship.
- Vague subsection headings (e.g., Section 5.1) and mistranslations (e.g., Section 2.2 title). Conduct full-text systematic revision.
- Conclusions require refinement.
The language expression lacks conciseness and clarity, and the results and discussions are intertwined, making it difficult to understand.
Author Response
Comments 1: The title of the article lacks conciseness, and it is recommended to revise it.
Response 1: Thank you for pointing this out, we agree with this comment. We have revised the title of the article according to your suggestions (Line 103)
Comments 2: Table 1 contains unit labeling errors. Systematically verify measurement units in all tables.
Response 2: Thank you for pointing it out. We have corrected the error in the article.(Table 1)
Comments 3: Figures and tables fail to meet academic standards (incomplete captions, insufficient logical coherence)
Response 3: We agree with this comment. We have redrawn and combined the figures involved in the article. ( Figure 1、Figure 2、Figure 3、Figure 5、Table 1)
Comments 4: Redundant descriptions in experimental procedures (e.g., Section 2.2). Streamline language and supplement critical parameters
Response 4: Thank you for pointing it out. We have reorganized this section according to your suggestions and supplemented the missing parameters. (Line103 to Line 150)
Comments 5: Ambiguous demarcation between two material preparation stages. Suggest separate subsections.
Response 5: Thank you for pointing it out. We have reorganized the experimental procedures according to your suggestions. (Line103 to Line 150)
Comments 6: Rename "Chemical Analysis" subsection to "FTIR Analysis" for accuracy
Response 6: Thank you for pointing it out. We have revised this subheading according to your suggestions。(Line194)
Comments 7: Optical microscopy mentioned in morphological analysis lacks corresponding result images. Add relevant images or remove unrelated method descriptions
Response 7: Thank you for pointing it out. Based on your suggestions, after careful consideration, we believe that the results obtained from the optical microscope cannot explain the issues in the article. Therefore, we have decided to delete this part of the results to avoid making the article overly lengthy and cumbersome.
Comments 8: Morphological Analysis (e.g., Section 3.1): Current optical micrographs only show single samples - insufficient for comparing three mixing methods.
Response 8: Thank you for pointing it out. According to your suggestions, we have reanalyzed the results of the optical microscope. We believe that these results are insufficient to illustrate the problem, so we have decided to delete this part of the results.
Comments 9: Water absorption discussion (3.4) contains divergent arguments. Focus on key data and remove redundant descriptions.
Response 9: Thank you for pointing it out. We have reanalyzed this part of the results according to your suggestions, focusing on the data and deleting the redundant parts. (Line 324 to Line 346)
Comments 10: Weak correlation between volume loss and density analysis (3.2). If merging discussions, supplement evidence clarifying their intrinsic relationship
Response 10: Thank you for pointing it out. According to your suggestions, we have supplemented the analysis of the correlation between the shrinkage rateand the density. (Line 287 to Line 296)
Comments 11: Vague subsection headings (e.g., Section 5.1) and mistranslations (e.g., Section 2.2 title). Conduct full-text systematic revision.
Response 11: Thank you for pointing it out. We have revised the subheadings throughout the full text according to your suggestions.
Comments 12: Conclusions require refinement
Response 12: Thank you for pointing it out. We have re-summarized the conclusion of the article according to your suggestions. (Line 460 to Line 472)
Thank you again for your guidance on improving this article! If you have any other suggestions for revision, we will definitely do our best to cooperate and refine it.
*Note: The revised parts are highlighted in yellow in the manuscript.
Reviewer 2 Report
Comments and Suggestions for Authors
In this study, the authors investigated the impact of different mixing methods of fungi and substrate on the mechanical and physicochemical properties of mycelium composites. The research is well-designed, and the findings hold potential for practical applications. However, the manuscript is poorly structured and often confusing, making it difficult to follow the results and conclusions drawn by the authors.
To improve clarity and coherence, the authors should consider addressing the following issues:
At the beginning of the introduction, provide a more detailed discussion on the significance of mycelium-based composite materials. Explain their applications and highlight their advantages over conventional materials.
Justify the selection of Ganoderma sichuanense and soybean straw as the fungal species and substrate. Provide an overview of their previous applications in related studies, citing relevant literature to support their suitability for this research.
When stating the research objective, specify the type of mycelium composite developed in this study, including its composition. Revise the sentence in Lines 73–79 to improve readability, as it is too long and difficult to follow
Line 85 - Before mentioning PDA, provide its full name followed by the abbreviation in parentheses. Additionally, include the manufacturer's details.
Line 96 In the super-clean workbench - Remove this throughout the text, as it is implicitly understood when working with microorganisms.
Lines 98-99, 112-114 - Specify the exact conditions under which the fungus was cultivated, including temperature, humidity, incubation time.
Lines 100-104 - I assume that the substrate was prepared identically for all three inoculations. Move this section to the 'Substrate Preparation' part of the manuscript
Lines 173-208 - Describe methods in a precise and scientific manner. Use clear and standardized terminology,
3.1 Morphological analysis - Rewrite this section to improve clarity, ensuring that the results are presented in a logical and structured manner. Begin with describing the findings shown in Figure 3, followed by those in Figure 2, without discussing or interpreting the results. Avoid subjective statements and maintain a clear, objective presentation of the data. Remove the discussion-like content found in Lines 290–29.
Avoid using sentences like Mycelium composite materials with a high density will still not be people's first choice
In some sections of the results, the findings are discussed. It is recommended to separate the results from the discussion to maintain clarity. Present the results objectively in one section, and reserve the interpretation and analysis for the discussion section.
Author Response
Comments 1: At the beginning of the introduction, provide a more detailed discussion on the significance of mycelium-based composite materials. Explain their applications and highlight their advantages over conventional materials.
Response 1: Thank you for pointing it out. We have added more content regarding the significance and advantages of mycelium-based composite materials according to your suggestions.(Line 29 to Line 33)
Comments 2: Justify the selection of Ganoderma sichuanense and soybean straw as the fungal species and substrate. Provide an overview of their previous applications in related studies, citing relevant literature to support their suitability for this research.
Response 2: Thank you for pointing it out. We have included the reasons for choosing Ganoderma sichuanase and soybean straws in the article according to your suggestions(Line 81 to Line 89)
Comments 3: Line 85 - Before mentioning PDA, provide its full name followed by the abbreviation in parentheses. Additionally, include the manufacturer's details.
Response 3: Thank you for pointing it out. We have added specific information to the article(Line 94)
Comments 4: Line 96 In the super-clean workbench - Remove this throughout the text, as it is implicitly understood when working with microorganisms.
Response 4: Thank you for pointing it out. We have made the deletions according to your suggestions.
Comments 5: Lines 98-99, 112-114 - Specify the exact conditions under which the fungus was cultivated, including temperature, humidity, incubation time.
Response 5: Thank you for pointing it out. We have supplemented the conditions for fungal cultivation in Section 2.2 according to your suggestions.
Comments 6: Lines 100-104 - I assume that the substrate was prepared identically for all three inoculations. Move this section to the 'Substrate Preparation' part of the manuscript
Response 6: Thank you for pointing it out. We have made adjustments to the content of Section 2.2 according to your suggestions.
Comments 7: Lines 173-208 - Describe methods in a precise and scientific manner. Use clear and standardized terminology,
Response 7: Thank you for pointing it out. We have revised the content of Section 2.3 according to your suggestions.
Comments 8: 3.1 Morphological analysis - Rewrite this section to improve clarity, ensuring that the results are presented in a logical and structured manner. Begin with describing the findings shown in Figure 3, followed by those in Figure 2, without discussing or interpreting the results. Avoid subjective statements and maintain a clear, objective presentation of the data. Remove the discussion-like content found in Lines 290–29.
Response 8: Thank you for pointing it out. We have reorganized the results of Section 3.1 according to your suggestions and deleted the redundant discussions.
Comments 9: Avoid using sentences like Mycelium composite materials with a high density will still not be people's first choice.
Response 9: Thank you for pointing it out. We have deleted this sentence according to your suggestion.
Comments 10: In some sections of the results, the findings are discussed. It is recommended to separate the results from the discussion to maintain clarity. Present the results objectively in one section, and reserve the interpretation and analysis for the discussion section.
Response 10: Thank you for pointing it out. We have re-summarized both the Discussion and Results sections according to your suggestions. (Lines 415 to Lines 472.)
Comments 11: In some sections of the results, the findings are discussed. It is recommended to separate the results from the discussion to maintain clarity. Present the results objectively in one section, and reserve the interpretation and analysis for the discussion section.
Response 11: Thank you for pointing it out. Through the analysis of density and elastic modulus values in the Ashby plot, the mycelium-based packaging composite material was identified as falling within the "foam" category. Fourier Transform Infrared (FTIR) spectroscopy analysis revealed that, in addition to the starch and proteins naturally present in wheat bran, the matrix composition of this composite material primarily contains varying amounts of macromolecules such as lignin, cellulose, and hemicellulose, along with chitin, proteins, and β-glucans.
Thank you again for your guidance on improving this article! If you have any other suggestions for revision, we will definitely do our best to cooperate and refine it.
*Note: The revised parts are highlighted in yellow in the manuscript.
Reviewer 3 Report
Comments and Suggestions for Authors
The study addresses an important topic in sustainable materials, specifically mycelium-based composites, which have potential applications in biodegradable packaging, insulation, and structural materials.
Mycelium growth varies depending on environmental factors (temperature, humidity, oxygen levels). How can the inconsistencies in composite properties be minimized?
The study acknowledges water absorption as a major drawback but does not explore potential surface treatments(e.g., hydrophobic coatings or heat pressing) to enhance moisture resistance.
Figures 2 and 4 (Material States and SEM Images)
- These figures are useful but lack clear labels to help readers quickly understand what is being compared.
- Adding arrows or markers to highlight key differences in mycelial coverage and microstructure would improve clarity.
- The significance of differences (p-values) is mentioned, but statistical tests are not clearly described in the Methods section.
- It would be helpful to briefly state what statistical method was used (e.g., ANOVA, t-test).
The discussion section mainly summarizes results but lacks a strong comparison with previous studies.
The conclusion should provide strong suggestion for future study such as: screening novel fungal strains or substrate composition to optimize material properties.
Comments on the Quality of English LanguageThe quality of English does not limit my understanding
Author Response
Comments 1: Mycelium growth varies depending on environmental factors (temperature, humidity, oxygen levels). How can the inconsistencies in composite properties be minimized?
Response 1: Thank you for pointing it out.Since mycelium-based composite materials are biological substances with inherent biological properties that cannot be fully controlled or artificially manipulated, our approach must involve strict adherence to experimental protocols and continuous monitoring of cultivation conditions to minimize experimental deviations.
Comments 2: The study acknowledges water absorption as a major drawback but does not explore potential surface treatments (e.g., hydrophobic coatings or heat pressing) to enhance moisture resistance.
Response 2: Thank you for pointing it out. We fully acknowledge that the material's water absorption constitutes an inherent limitation. While the experimental design was intentionally formulated to utilize water absorption capacity as a critical parameter for evaluating three mixing methods, further elaboration on moisture reduction strategies would constitute a divergence from this study's primary research objectives. This strategic decision precluded comprehensive analysis of hygroscopic modification in the current work, notwithstanding our recognition of its significance as a valuable research avenue. Subsequent investigations will specifically address the moisture regulation mechanisms through dedicated investigation.
Comments 3: Figures 2 and 4 (Material States and SEM Images). These figures are useful but lack clear labels to help readers quickly understand what is being compared.Adding arrows or markers to highlight key differences in mycelial coverage and microstructure would improve clarity.
Response 3: Thank you for pointing it out. We have recombined the macroscopic and microscopic images of the materials to make them clearer. (Figure 1、Figure 2)
Comments 4: The significance of differences (p-values) is mentioned, but statistical tests are not clearly described in the Methods section. It would be helpful to briefly state what statistical method was used (e.g., ANOVA, t-test).
Response 4: Thank you for pointing it out. We have added data analysis to the Methods section according to your suggestion. (Lines 205 to Lines 210.)
Comments 5: The discussion section mainly summarizes results but lacks a strong comparison with previous studies.
Response 5: Thank you for pointing it out. We have revised the Discussion section according to your suggestions and added comparisons with other literatures. (Lines 415 to Lines 458.)
Comments 6: The conclusion should provide strong suggestion for future study such as: screening novel fungal strains or substrate composition to optimize material properties.
Response 6: Thank you for pointing it out. We have revised the Conclusion section according to your suggestions and added comments on future research.
Thank you again for your guidance on improving this article! If you have any other suggestions for revision, we will definitely do our best to cooperate and refine it.
*Note: The revised parts are highlighted in yellow in the manuscript.
Reviewer 4 Report
Comments and Suggestions for Authors
Review
for the article entitled "The influences of different mixing methods of fungi and substrate on the mechanical properties and physicochemical properties of mycelium composites"
The authors investigated the effect of different methods of fungal-substrate mixing on the mechanical properties (mechanical strength, toughness and modulus of elasticity) and physico-chemical properties (density, water absorption, chemical composition, substance stability) of mycelial composites. The present article is in line with the scope of the journal Biology and will serve as a basis for optimising the preparation and production of mycelial composites. It is therefore recommended for publication in Biology. However, the authors are encouraged to revise the manuscript with respect to the following points:
Point 1: Page 4, lines 160-168 and Figure 1 should be moved to the Results section.
Point 2: The "Methods" section should include a subsection on "Statistical analysis".
Point 3: In the "Results" section, the results obtained by the authors are interspersed with literature data in almost every paragraph, which makes it very difficult to read the text of the article. It is necessary to first describe the results obtained by the authors, including tables and figures, and then discuss them with the literature data.
Author Response
Comments 1: Page 4, lines 160-168 and Figure 1 should be moved to the Results section
Response 1: Thank you for pointing it out. We have adjusted the relevant figures and results according to your suggestions.(Figure 2)
Comments 2: The "Methods" section should include a subsection on "Statistical analysis".
Response 2: Thank you for pointing it out. We have added “Statistical analysis” section according to your suggestions. (Lines 206 to Lines 210.)
Comments 3: In the "Results" section, the results obtained by the authors are interspersed with literature data in almost every paragraph, which makes it very difficult to read the text of the article. It is necessary to first describe the results obtained by the authors, including tables and figures, and then discuss them with the literature data.
Response 3: Thank you for pointing it out. We have made adjustments according to your suggestions, following the order of first presenting the obtained results and then discussing them in comparison with the literature.( The revised parts have been highlighted with yellow in the text.)
Thank you again for your guidance on improving this article! If you have any other suggestions for revision, we will definitely do our best to cooperate and refine it.
*Note: The revised parts are highlighted in yellow in the manuscript.
Round 2
Reviewer 1 Report
Comments and Suggestions for Authors
The manuscript can be accepted now
Author Response
We sincerely appreciate the reviewers' professional critiques and valuable recommendations, which have enhanced both the technical accuracy and presentation clarity of the manuscript!
Reviewer 2 Report
Comments and Suggestions for Authors
Revise the sentence in Lines 73–79 to improve readability, as it is too long and difficult to follow
Lines 168-204 - Describe methods in a precise and scientific manner. Use clear and standardized terminology,
Author Response
Comments 1:Revise the sentence in Lines 73–79 to improve readability, as it is too long and difficult to follow
Response 1: Thank you for pointing this out, according to your suggestion, we have made necessary deletion and rewriting of lines 73 – 79 to make it easier to read.
Comments 2:Lines 168-204 - Describe methods in a precise and scientific manner. Use clear and standardized terminology
Response 2: Thank you for pointing this out,according to your suggestions, we have made reasonable modifications to the lines 168-204 - describe methods section by querying the information to make it look scientific and standard.
Thank you again for your guidance on improving this article! If you have any other suggestions for revision, we will definitely do our best to cooperate and refine it.
*Note: The revised parts are highlighted in blue in the manuscript.
Reviewer 3 Report
Comments and Suggestions for Authors
The authors have improved their manuscript
Author Response

(The authors gave the same response as above.)
